# MeMo: Meaningful, Modular Controllers Via Information Bottlenecks

## Abstract

Robots are often built from standardized assemblies, (e.g. arms, legs, or fingers), but each robot must be trained from scratch to control all the actuators of all the parts together. In this paper we demonstrate a new approach that takes a single robot and its controller as input and produces a set of modular controllers for each of these assemblies such that when a new robot is built from the same parts, its control can be quickly learned by reusing the modular controllers. We achieve this with a framework called MeMo which learns (Me)aningful, (Mo)dular controllers. Specifically, MeMo pretrains a modular architecture that assigns separate neural networks to physical substructures and uses an information bottleneck to learn an appropriate division of control information between the modules. We benchmark our framework in locomotion and grasping environments on challenging simple to complex robot morphology transfer. We also show that the modules help in task transfer. On both structure and task transfer, MeMo achieves improved training efficiency to pretrained graph neural network baselines. In particular, MeMo significantly improves training efficiency on structure transfer, often achieving 2x the training efficiency of the strongest baseline.

## 1 Introduction

Consider the following scenario: A roboticist is designing a robot with 6 legs, such as the one seen in the left image of Fig. 1, and has trained a standard neural network controller with deep reinforcement learning (RL) to control the actuators circled in green. However, after more testing, they realize that the design of the robot needs to be extended with another pair of legs to support the desired amount of weight. Even though the new 8 leg robot is still composed of the same standard assemblies, the roboticist is unable to reuse any part of the 6 leg robot's controller. While many works (Huang et al., 2020; Kurin et al., 2021; Gupta et al., 2022) have studied structure transfer, or transferring neural network controllers to different robot morphologies, these works take a purely data-driven approach of training a universal controller on a dataset representative of the diversity and complexity of robots seen in testing. In contrast, we desire to learn transferable controllers from only a single robot and environment, obviating the requirement for a substantial training dataset and resources to perform multi-task RL. Our experiments demonstrate that state-of-the-art approaches for transferring control to environments with incompatible state-action spaces struggle to generalize in this highly data-scarce setting.

Motivated by the above scenario, we propose a framework, MeMo, for pretraining (Me)aningful (Mo)dular controllers that enable transfer from a single robot to variants with different dimensionalities. Due to the difficulty of one-shot transfer, we focus on transferring to variants of the training robot to ensure that the dynamics inferred during training are still useful in the transfer environment, an assumption which is unlikely to hold if the transfer robot has a significantly different global morphology. Yet, without any additional knowledge, transfer from only a single robot and task is still a daunting challenge. The key insight MeMo leverages is that a robot is built from assemblies of individual components, such as the leg of a walking robot or the arm of a claw robot. These assemblies are specified by a domain expert who is able to account for the constraints imposed by the robot's hardware implementation in their specification. Given this information, MeMo learns controllers responsible for coordinating the individual actuators that comprise a given assembly, and these assembly-specific controllers, or modules, are coordinated by a higher-level master controller. As we are able to reuse the modules when transferring to a robot built from the same assemblies, the

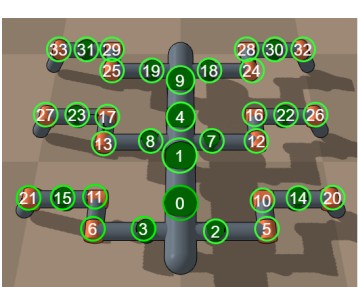 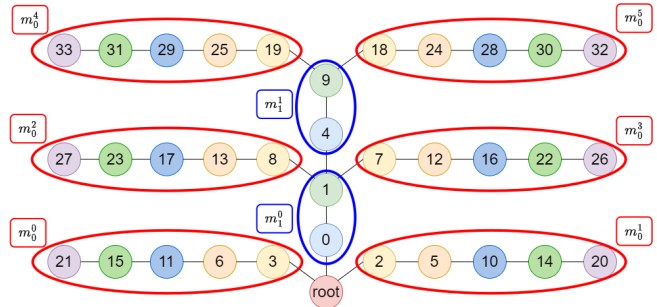

Figure 1: **Graph Structure and Neural Network Modules of the 6 Leg Centipede. Left:** The robot's joints are labeled numerically and circled. **Right:** The joints form the nodes and the links are the edges. The subset of joints that form each leg module are circled in red, while those that comprise each body module are circled in blue. Neural network modules are denoted as $m_k^i$, where $k$ refers to the type, e.g. all leg modules are type 0, and $i$ denotes different instances of the same module type.

problem of learning a controller for a different morphology boils down to learning the coordination mechanics among assemblies, rather than having to coordinate at the granular level of individual joints. Returning to the 6 leg robot seen in Fig. 1, we see that the robot is comprised of multiple "leg" and "body" assemblies that are circled in red and blue respectively in the right image. Module parameters are shared between assemblies of the same type, providing multiple training instances that help our modules generalize. After training the modules with MeMo, the "leg" and the "body" modules can then be reused to speed up the training of a different robot's controller, such as an 8 leg robot.

To achieve this improved training efficiency, a key challenge is to balance the labor between the master controller and the modules. In one direction, to prevent the modules from becoming too robot-specialized, we introduce information asymmetry into our architecture, where the modules are limited to seeing the local observations of the actuators that belong in the module. In the other direction, controlling the assembly through the module must be simpler than controlling the assembly directly, since otherwise there is no benefit to this new architecture. This is achieved by a new policy learning scheme that imposes an information bottleneck between the master controller and the modules.

To evaluate the transferability of the learned module, we apply MeMo in locomotion and grasping domains. Specifically, we design two types of transfer: generalizing to more complex robot structures and to more difficult tasks. When transferring model weights from a simpler agent, we show that MeMo significantly improves the sample efficiency of performing RL on more complex agents. We compare our approach with NerveNet, an alternative approach for one-shot structure transfer (Wang et al., 2018; Blake et al., 2021). Our experiments show that MeMo either exceeds or matches NerveNet's training efficiency during transfer, as NerveNet's message-passing policy is prone to overfitting to the original robot structure during pretraining.

## 2 RELATED WORK

Our work relates to modular controllers, information bottlenecks, and structure transfer. Much prior work in modular robotics only consider individual actuators as modules, while our work enables domain experts to specify assemblies of one or more actuators that align with their hardware constraints. Related works in multi-robot and hierarchical RL, discussed further in Appendix A.8, share some similarities with our work in the use of a hierarchical architecture or an information bottleneck. While most work in structure transfer assume the availability of a multi-task dataset of robots and environments, we focus on transferring policies learned on a single robot and environment.

**Modular Controllers.** Our work relates to prior works that train modular policies for robot designs. Devin et al. (2017) learns neural network policies that are decomposed into "task-specific" and "robot-specific" modules and performs zero-shot transfer to unseen task and robot-specific module combinations. Huang et al. (2020) coordinates modular policies shared among all actuators via message passing. Whitman et al. (2021) uses a GNN to internally coordinate between part-specific

nodes and share module parameters between nodes corresponding to the same part. In both of these works, the training dataset consists of robot designs across a variety of tasks, and the test dataset consists of unseen designs and tasks. Whitman & Choset (2022) combines RL and IL to train a modular, message-passing policy more efficiently. Pathak et al. (2019) proposes the Dynamic Graph Network to control self-assembling agents, consisting of modules that are shared across agents.

**Information Bottleneck.** Our method involves imposing an information bottleneck between the master controller and the modules in order to ensure that the modules learn meaningful behavior. Recent works (Igl et al., 2019; Goyal et al., 2020) have used the Variational Information Bottleneck (Alemi et al., 2017) in RL and HRL. Goyal et al. (2020) decompose a policy into a mixture of primitives that represent modular behaviors, and each modules' access to the full state vector is limited through a KL-divergence term. In supervised learning, regularization techniques involving noise are often used, such as dropout (Srivastava et al., 2014), batch normalization (Ioffe & Szegedy, 2015), and augmenting the inputs with noise (Devries & Taylor, 2017). Another method of regularization is enforcing sparsity (Cranmer et al., 2020; Gupta et al., 2023) in the model. For example, Cranmer et al. (2020) makes GNN messages sparser through L1 regularization. Gupta et al. (2023) learns extremely sparse intermediate representations while still maintaining a high end-to-end accuracy.

**Structure Transfer.** In the hierarchical RL setting, Hejna et al. (2020) proposes learning high-level policies that represent long-horizon behavior via imitation-learning and improve sample efficiency when transferred from simple to complex agents. Liu et al. (2022) transfers policies to robots with significantly different kinematics and morphology by defining a continuous evolution from the source to the target robot. Previous works explore the use of message-passing policy architectures to generalize across morphologies (Wang et al., 2018; Blake et al., 2021; Huang et al., 2020). In the multi-task setting Kurin et al. (2021) proposes Transformers as policy representations that ignore morphological knowledge, removing the need for multi-hop communication. Gupta et al. (2022) scales Transformer-based policies to larger and more diverse datasets of robots.

## 3  METHOD

In this section, we describe in detail our approach MeMo, an algorithm for pretraining reusable control modules. We formulate control problems as a finite-horizon discounted Markov decision process, defined by a state or observation space $\mathcal{S}$ and action space $\mathcal{A}$. The agent's stochastic policy for interacting with the environment is described as $\pi_\theta(a^\tau \mid s^\tau)$, where $s^\tau \in \mathcal{S}$, $a^\tau \in \mathcal{A}$, and $\theta$ are the policy function's parameters. The environment produces a reward $r(s^\tau,\ a^\tau)$ for the agent. The agent's objective is to maximize the expected discounted return $J(\theta) = \mathbb{E}[\sum_{\tau=0}^{\infty} \gamma^\tau r(s^\tau, a^\tau)]$, where $\gamma \in [0, 1)$ is the discount factor.

### 3.1  HIERARCHICAL ARCHITECTURE

Formally, we assume that we are given a partitioning $\mathcal{P}$ of an agent's joints $j_{0,...,N-1}$. We design a hierarchical policy composed of a master controller $M$ that outputs intermediate signals to neural network modules that decode actions. Each element of the partition, e.g. a subset of actuators, is a module instance $i$ of type $k$, which we denote as $m_k^i$. In total, there are $|\mathcal{P}|$ modules. Modules of the same type $k$ share the module parameters, yet each instance will receive a different message from $M$.

Specifically, as shown in Fig. 2, the hierarchical architecture starts by executing the master controller $M$, which takes the full observation vector $s^\tau$ as input. The full observation vector $s^\tau$ includes both global observations about the agent and local observations of each actuator. The global observations consist of the agent's global position, orientation, and velocity, while the local observations are composed of joint angle, joint velocity, and local relative position and orientation in the corresponding module's frame. Given the full observation vector $s^\tau$, $M$ outputs a latent vector $H$ of length $|\mathcal{P}| \cdot D$, where $D$ is the size of the embedding sent to each module.

The latent vector $H$ is then split into $|\mathcal{P}|$ segments of size $D$ and fed to modules. As shown in Fig. 3, a module itself consists of a MLP for each actuator $j_n$. Each MLP takes as input $j_n$'s local features concatenated with the module's segment of the latent vector $H$ and outputs the mean value of the action applied to $j_n$.

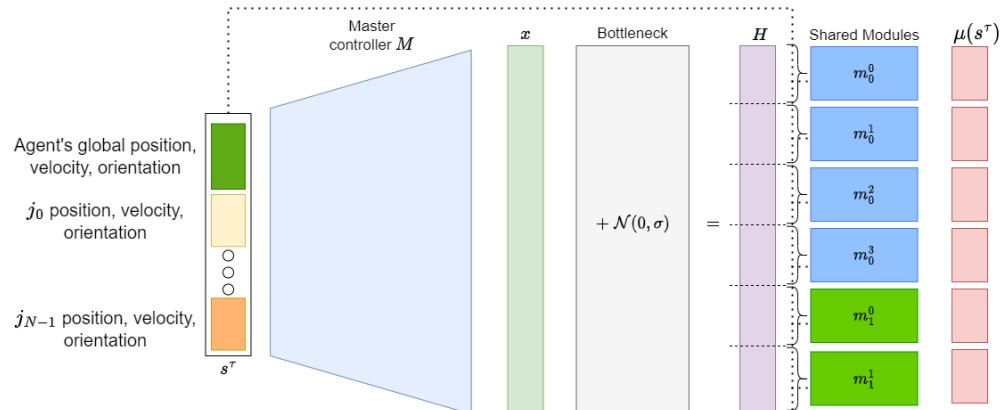

Figure 2: **Hierarchical Architecture with Information Bottleneck.** Our architecture consists of a higher-level master controller $M$ that outputs a hidden embedding, $x$. During imitation learning, Gaussian noise is added to $x$ to compute $H$. $H$ is split into signals that are passed into modules that output the mean of the action distribution. The dotted lines represent that in addition to $H$, the modules also take in subsets of the full observation vector corresponding to the state of the joints within the modules.

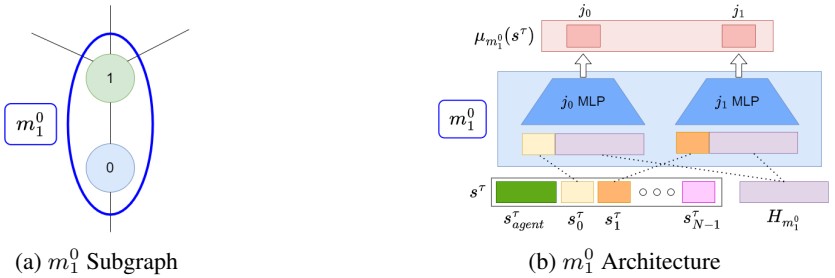

(a) $m_1^0$ Subgraph

(b) $m_1^0$ Architecture

Figure 3: **Module Subgraph and Architecture. Left:** Module $m_1^0$ is responsible for computing the mean actions of actuators 0 and 1. **Right:** A module consists of separate networks that compute each actuator's mean action. The inputs include the local observations of the actuator concatenated with the signal sent to the module it belongs to.

## 3.2 TRAINING PIPELINE

Despite the transferability of our hierarchical policy architecture, imposing an information bottleneck indeed increases the difficulty of training such a policy from scratch. Inspired by previous works that combine RL and IL (Chen et al., 2021; Zhu et al., 2019; Radosavovic et al., 2021) to accelerate RL training, we first train an expert MLP controller with RL and perform imitation learning (IL) to pretrain the modules. After transferring the modules to a new structure or task, we perform RL to retrain the master controller or finetune the architecture end-to-end. Our pipeline is summarized in Fig. 4. Ablation experiments (Section 4.3) show that pretraining modules with RL does not yield modules that improve sample efficiency on transfer.

### 3.2.1 REINFORCEMENT LEARNING

During the reinforcement learning stage, we train actor-critic controllers for each robot with proximal-policy optimization (PPO) (Schulman et al., 2017). The critic is a MLP, whereas the actor is a standard MLP when training the expert controller and the hierarchical architecture (Section 3.1) when transferring pretrained modules. As typical in PPO, we use Tanh nonlinearities and orthogonal initialization in all architectures. In PPO, agents iteratively sample trajectories based on the current policy and subsequently perform optimization on a surrogate objective that first-order approximates the natural gradient. The surrogate objective prevents unstable updates to the policy by clipping the

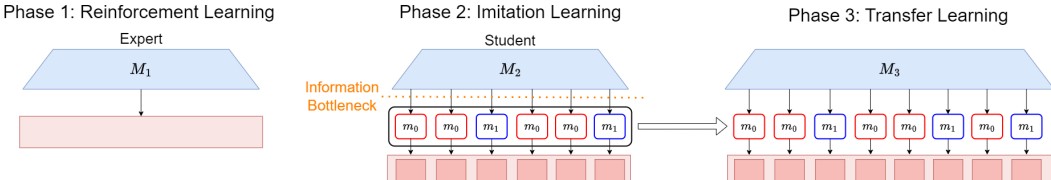

Figure 4: **Training Pipeline Overview.** In Phase 1, we first train an expert controller for the training robot using RL. In Phase 2, we pretrain modules with imitation learning while imposing an information bottleneck between the master and the modules. In Phase 3, we transfer the modules to a different context and retrain the master controller $M_3$.

probability ratio $r^\tau(\theta; \theta_{old}) = \pi_\theta(a^\tau \mid s^\tau) / \pi_{\theta_{old}}(a^\tau \mid s^\tau)$. Optimizing the clipped objective is done with the policy gradient (Sutton et al., 1999).

### 3.2.2 IMITATION LEARNING WITH INFORMATION BOTTLENECK

The goal of imitation learning (IL) is to train modules that are transferable to a variety of contexts, such as different robot structures and tasks. This requires the modules to represent meaningful behavior, which we enforce by imposing an information bottleneck on the master controller while training the hierarchical architecture end-to-end. We assume access to a given robot's MLP controller, trained with RL as described in Section 3.2.1. As such, we use DAgger (Ross et al., 2011) to train a hierarchical architecture $\theta$ end-to-end. At each iteration $i$ of DAgger, we sample a trajectory $\mathcal{D}_i$ from $\theta$ with maximum $T$ steps. The expert provides the correct action to each $s \in \mathcal{D}_i$, and $\mathcal{D}_i$ is aggregated into the full dataset $\mathcal{D} = \{(s, a)\}$. The supervised learning objective is to minimize the loss $\mathcal{L} = -\mathbb{E}_{s \sim \mathcal{D}}[\log \pi_\theta(a \mid s)]$.

Our information bottleneck involves adding Gaussian noise sampled from the distribution $\mathcal{N}(0, \sigma)$ to each dimension of the master controller output, as shown in Fig. 2. $\sigma$ is a hyperparameter to control the scale of the noise, and we find $\sigma = 1$ works well in practice. Intuitively, the addition of noise encourages the modules to rely less on the master controller's signal. Furthermore, we find that the bottleneck improves the robustness of the model to unstable policy updates, allowing us to use smaller batch sizes in RL when transferring pretrained modules. We provide ablation studies (Section 4.3) and analysis (Section 4.4) of our Gaussian noise bottleneck, demonstrating that it is essential for training modules that improve RL sample efficiency.

## 4 EXPERIMENTS

With our experiments, we seek to answer four questions. 1) Do the modules produced by MeMo generalize when transferred to different robot morphologies and tasks? 2) When pretraining modular controllers with imitation learning, does the Gaussian noise information bottleneck on the master signal help? 3) In the pretraining phase, why do we use imitation learning rather than applying the bottleneck in reinforcement learning? 4) How does applying the bottleneck yield better representations of the actuator space? We answer question 1) in Sections 4.1 and 4.2, 2) and 3) in Section 4.3, and 4) in Section 4.4.

### 4.1 TRANSFER LEARNING

We benchmark our framework on two types of transfer: structure and task transfer. While our framework is designed primarily for structure transfer, we use task transfer experiments as an additional means of evaluating the quality of the learned representations. For the locomotion experiments, we perform experiments on the tasks introduced in RoboGrammar (Zhao et al., 2020) with training statistics computed as the average reward across 3 runs, with standard deviations indicated by shaded areas. For the grasping domain, we construct object-grasping tasks using the DiffRedMax simulator (Xu et al., 2021) and compute training statistics as the average reward across 5 runs. Additional details on the reward functions used are in Appendix A.5. We visualize train and test robot morphologies for structure transfer in Fig. 5 and the train and test tasks for task transfer in Fig. 13.

**Locomotion.** We design three structure transfer tasks in the locomotion domain, in which the goal is to move as far as possible while maintaining the robot's initial orientation. The starting morphologies are the 6 leg centipede robot, the 6 leg worm robot, and the 6 leg hybrid. The 6 to 12 leg centipede transfer demonstrates scalability to transfer robots with many more modules than seen in training. The 6 to 10 leg worm shows that MeMo generalizes with only 1-2 instances of the same module seen in training. The 6 and 10 leg hybrid robots involve three types of modules, demonstrating scalability to more complex training robots. For task transfer, we transfer policy weights pretrained on a 6 leg centipede locomoting over the Frozen Terrain to three different terrains that feature obstacles or climbing.

**Grasping.** In grasping, the goal is to grasp and lift an object as high as possible. We design a grasping robot consisting of an arm that lifts a claw grasping a cube. The structure transfer is from a 4 finger to a 5 finger claw. For task transfer, we transfer policies trained to control the 4 finger claw grasping a cube to the same robot grasping a sphere of similar size and weight.

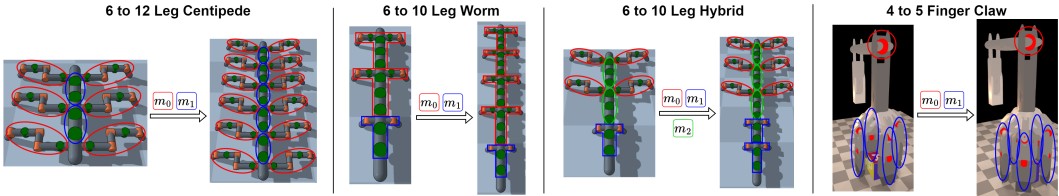

Figure 5: **Structure Transfer Tasks. Left:** Transfer "leg" and "body" modules from a 6 to a 12 leg centipede. **Left Middle:** Transfer "body" and "head" modules from a 6 to a 10 leg worm. **Right Middle:** Transfer "leg," "head," and "body" modules from a 6 to a 10 leg hybrid. **Right:** Transfer "arm" and "finger" modules from a 4 to a 5 finger claw.

**Baselines.** We compare MeMo with several baselines. In particular, we compare the generalization ability of MeMo to pretrained NerveNet (Wang et al., 2018; Blake et al., 2021) baselines, which have been evaluated on transfer from a single robot and task. NerveNet takes input the underlying graph structure of the agent, in which the nodes are actuators and edges are body parts. The graph structures of the train morphologies are detailed in Appendix A.10. Full training details and state space descriptions are included in Appendix A.2 and A.3 respectively.

- **RL (MLP)**: For structure transfer, due to the change in the observation space, we train a 2 layer MLP policy from scratch with RL. In task transfer, we use a MLP pretrained with RL on the original task and finetune it on the test task. For a fair comparison, we use the same architecture size as the hierarchical architecture's master controller and replace the modules with a linear layer decoder.

- **RL (Hierarchical)**: For structure transfer, we train the hierarchical architecture, discussed in Section 3.1, from scratch with RL. In task transfer, we use a hierarchical architecture pretrained with RL on the training task and finetune both the modules and the master controller on the test task. The inclusion of this baseline allows us to isolate the effect of the hierarchical architecture from the pretraining and bottleneck components of MeMo.

- **Pretrained NerveNet-Conv**: We use the NerveNet network architecture proposed by Wang et al. (2018), consisting of an input network $F_{in}$ for encoding observations, a message function $M$, an update network $U$, and an output network $F_{out}$ for decoding. As in Wang et al. (2018), $F_{in}$ and $F_{out}$ are MLPs. In the convolutional (Kipf & Welling, 2017) variant, $M$ is the identity function and $U$ is a weight matrix. During RL, we fix $F_{out}$ in a similar manner as fixing the modules in MeMo, which improves NerveNet-Conv's performance.

- **Pretrained NerveNet-Snowflake:** Snowflake (Blake et al., 2021) is a state-of-the-art approach for training GNN policies that scale to high-dimensional continuous control. Their method involves fixing parts of the NerveNet architecture to prevent overfitting during PPO. Empirically, they find that fixing $\{F_{in}, M, F_{out}\}$ results in the best performance on MuJoCo tasks. We follow the same parameter fixing as Snowflake. As in Snowflake, we parameterize $F_{in}$ and $F_{out}$ as MLPs and the update function $U$ as a GRU. We use a weight matrix for $M$.

## 4.2 RESULTS

The generalization ability of MeMo on structure transfer is shown in Fig. 6. On all structure transfer tasks, MeMo outperforms both GNN baselines. On the 12 leg centipede and the 10 leg hybrid, not only is MeMo 2× more sample efficient than the best baseline, but it also converges to controllers with significantly better performance than any baseline. On the 5 finger claw, MeMo achieves a similarly performing policy as the baseline while only using one-third of total timesteps. On the 10 leg worm, MeMo outperforms all baselines in terms of training efficiency and achieves a comparable final performance as NerveNet-Conv. We note that the worm transfer task is easier for GNN models, because the coordination of the shorter legs and body joints is naturally captured with multi-hop communication.

The results of MeMo on task transfer are shown in Fig. 7. As transferring from the Frozen to the Ridged, Gap, and Stepped Terrains requires the robot to overcome obstacles unseen in the Frozen Terrain, we load the pretrained master controller and finetune MeMo end-to-end. Results (Fig. 7) show that MeMo achieves improved training efficiency on all test tasks compared to pretrained MLP and hierarchical architectures. MeMo achieves comparable performance on the Ridged and Gap Terrains and outperforms the NerveNet baselines on the Stepped Terrain, which requires the robot to climb up steps whereas the training terrain is flat. MeMo also has improved training efficiency and final performance in the grasping domain when transferring from grasping a cube to a sphere. The pretrained NerveNets struggle to coordinate the arm and claw components, resulting in high variance across different random seeds.

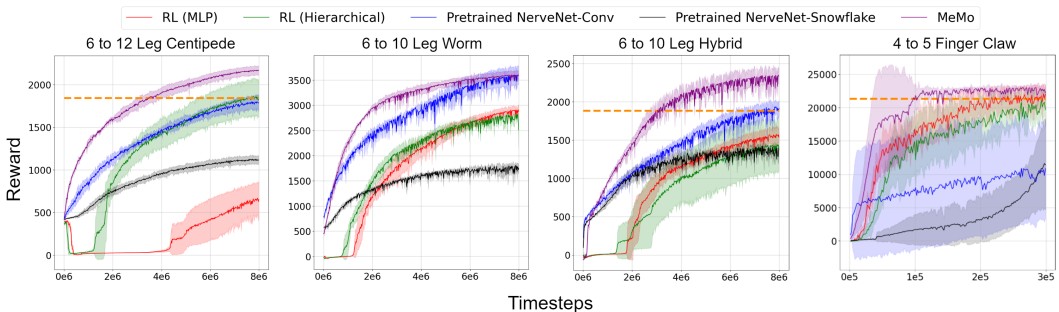

Figure 6: **Structure Transfer Results. Left:** 6 leg centipede to 12 leg centipede transfer on the Frozen Terrain. **Left Middle:** 6 leg worm to 10 leg worm transfer on the Frozen Terrain. **Right Middle:** 6 leg hybrid to 10 leg hybrid transfer on the Frozen Terrain. **Right:** 4 finger claw to 5 finger claw transfer on grasping a cube. The dashed orange line shows that the final performance of the closest baseline is achieved by MeMo within half or a third of the total number of timesteps.

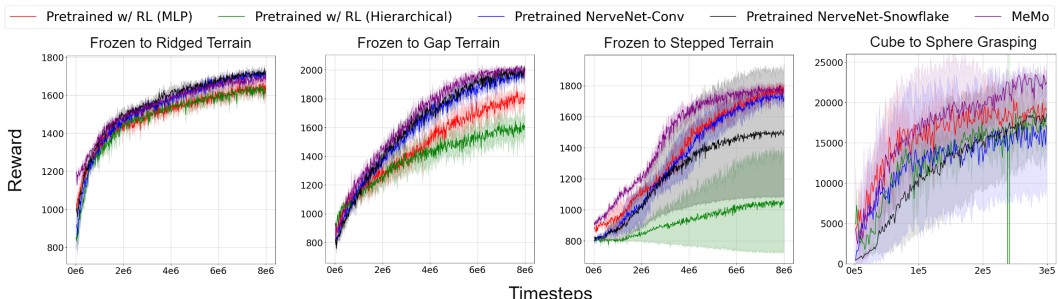

Figure 7: **Task Transfer Results. Left:** The first three plots show results on transferring from the 6 leg centipede walking over the Frozen Terrain to the same centipede walking over a terrain with ridges, a terrain with gaps, and a terrain with upward steps. **Right:** The last plot shows the transfer from a 4-finger claw grasping a cube to the same claw grasping a sphere. MeMo either has comparable training efficiency to the strongest baseline or outperforms all baselines.

### 4.3    ABLATION STUDY

#### 4.3.1    INFORMATION BOTTLENECK

The key to the success of MeMo is the introduced Gaussian noise information bottleneck, which encourages proper responsibility division among the pretrained master controller and modules, enabling the modules to improve training efficiency when reused. In this section, we conduct an ablation study to verify this technique. To do so, we experiment on a special setting, "transferring" the controller to the same robot structure and task, a 6 leg centipede traversing a Frozen Terrain. During transfer, we reuse and freeze the pretrained modules and retrain the master controller from scratch. With the pretrained modules from MeMo, the master controller will be retrained much more efficiently because it only needs to take partial responsibility for the control job. We compare our method to two baselines:

- **MeMo (no IB)**: We pretrain the hierarchical architecture end-to-end without an information bottleneck. This ablation is equivalent to MeMo without the bottleneck.

- **MeMo (L1):** During pretraining, we replace the Gaussian noise bottleneck with L1 regularization on the master controller's output that encourages sparsity in the master signal. We weigh the regularization term by a hyperparameter $w$ and report results with the best $w$.

In addition, we add the training curve of the **RL (Hierarchical)** as a reference. The ablation results are shown in Fig. 8a. From the plot, we can see that MeMo yields a significant improvement in training efficiency over all ablations.

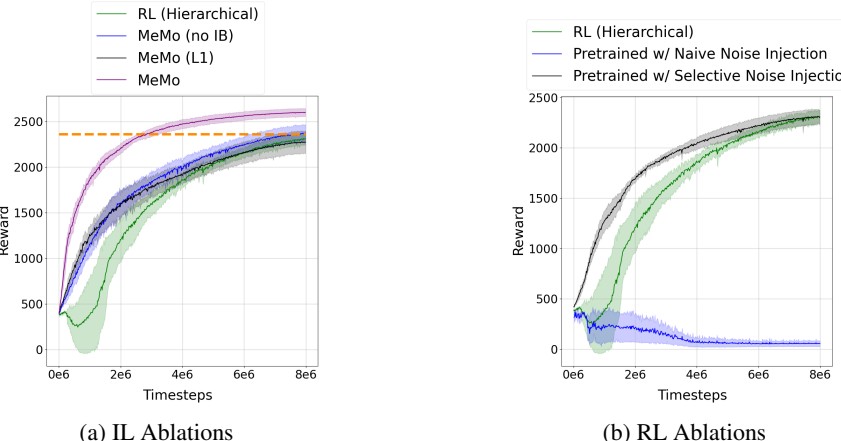

(a) IL Ablations                                              (b) RL Ablations

Figure 8: **Ablation Results. Left:** Our proposed bottleneck outperforms all other variants. The dashed orange line shows that the final performance of the closest baseline is achieved by MeMo in less than half of the total number of timesteps. **Right:** Pretraining modules with reinforcement learning results in the same performance as training the hierarchical architecture from scratch.

#### 4.3.2    IMITATION LEARNING

We now answer the second question of whether imitation learning is necessary to pretrain modules with a Gaussian noise information bottleneck. The results of using our bottleneck in reinforcement learning to pretrain modules is shown in Fig. 8b. Note that we refer to IL ablations as experiments where modules are first pretrained with imitation learning, and subsequently, the master controller is reinitialized and retrained with RL to test the improvement in sample efficiency. RL ablations involve the second RL phase, but the pretraining stage is done with RL as well. We find that the modules pretrained with RL result in the same performance as training the hierarchical architecture from scratch. We experiment with two methods of injecting noise during RL. The first is naively applying our Gaussian noise bottleneck to the master signal when sampling rollouts and computing policy gradients. For the second, we adopt the Selective Noise Injection (SNI) technique proposed by Igl et al. (2019) for applying regularization methods that have stochastic components in RL. SNI

stabilizes training by sampling rollouts deterministically and computing the policy gradient as a mixture of gradients from the deterministic and stochastic policies. However, even with SNI, the pretrained modules do not improve training efficiency.

## 4.4 ANALYSIS

We examine how imposing an information bottleneck forces the modules to learn a better representation of the actuator space. Intuitively, while the space of actuator signals is high-dimensional, the trajectories produced by a successful policy lie on a much lower-dimensional manifold, in which each dimension represents an individual skill that the policy has learned. We can measure the dimensionality of the modules' mapping by looking at the Jacobian matrix of the actuators with respect to the master signal. Consider a toy example in which the actuator space is 3-dimensional, yet the positions of the actuators vary along a straight line, which is a 1-dimensional manifold. The Jacobian of a well-trained module would only have one non-zero eigenvalue corresponding to the single direction of variance along the line. In a less contrived scenario, the trajectories outputted by a policy can likely be captured by a few dimensions of high variance corresponding to a small set of large eigenvalues in addition to a much larger set of dimensions of lower variance corresponding to relatively small eigenvalues.

We visualize this effect by 1) computing the Jacobians at the trajectory input states of a successful policy and 2) normalizing the eigenvalues of each Jacobian by its largest eigenvalue and plotting the resulting values in the [0, 1] range. We expect to see for a well-trained module that the normalized eigenvalue distribution is right skewed, with the distribution tail corresponding to the dimensions that matter most while the majority of the distribution's mass falls close to 0. Conversely, modules that do not produce a low-dimensional manifold would have eigenvalues that are similar values, resulting in the distribution's mass clustering close to 1. We verify this intuition by sampling 100 trajectories from an expert controller for the 6 leg centipede shown in Fig. 1. For the ablations of MeMo described in Section 4.3, we plot the normalized eigenvalues in Fig. 9. In the leftmost plot corresponding to a hierarchical architecture trained without any bottleneck, the majority of the values are close to 1. At the other extreme, with our Gaussian noise bottleneck, the values are highly clustered to the left, implying that most eigenvalues are much smaller than the biggest eigenvalue. We see that a lower noise scale has a less noticeable skew and that L1 regularization only results in a slightly skewed distribution.

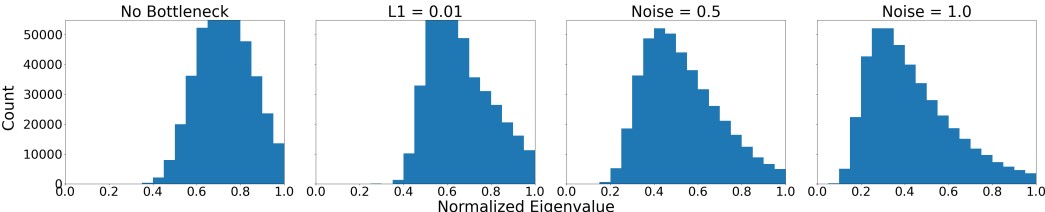

Figure 9: **Eigenvalue Distributions of Actuator-Master Jacobians.** We pretrain the hierarchical architecture with various bottlenecks and plot the normalized eigenvalues of Jacobian matrices computed for an expert's trajectories. With the white noise bottleneck, the distributions are much more skewed right, showing that the modules learn better representations of the actuator space.

## 5 CONCLUSION

In this paper, we propose a hierarchical, modular architecture for robot controllers, in which a higher-level master controller coordinates lower-level modules that control shared physical assemblies. We train the architecture end-to-end with an information bottleneck to ensure that the lower-level modules do not overrely on the master controller's signal. In locomotion and grasping environments, we demonstrate that our pretrained modules outperform GNN-based methods when transferring from simple to complex morphologies and to different tasks. We ablate components of MeMo and demonstrate that the entire framework is necessary to achieve these generalization benefits.

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

# A APPENDIX

## A.1 LIMITATIONS AND FUTURE WORK

We now discuss the limitations of our work and potential future directions. One limitation is that our experiments were conducted using only a single type of RL algorithm (PPO) and network architecture, MLPs with Tanh nonlinearities and orthogonal initialization. Future work would involve applying and adapting our framework to different policy optimization algorithms and architectures.

While we have demonstrated the potential of MeMo to be used for task transfer, such capabilities are inherently limited as our architecture does not explicitly encode the semantics of the task. For example, transferring the modules of a 6 leg robot walking over a Frozen Terrain to the same robot traversing a terrain with obstacles requires finetuning the architecture end-to-end, which may not always yield better results due to the instability of policy updates. Combining our framework with a mechanism for representing task semantics to enable transfer to more complex tasks is a promising direction for future work.

The scope of our work is limited to robots that are incrementally different from the starting robot, due to the difficulty of generalizing from a single robot and environment. One line of future work could involve adapting our training pipeline to multirobot and multitask settings, enabling our modules to capture a broader range of robot dynamics. When the dataset of robots grows larger, it can be expensive for a domain expert to manually provide labels on how a robot is decomposed into physical assemblies. However, the problem of learning reusable components bears similarity to the problem of abstraction learning studied in the programming languages community. Recent advances (Bowers et al., 2023; Cao et al., 2023) have made abstraction learning much more computationally efficient than in the past. We also see promise in adapting these techniques, which have been developed for programs, to robots that have an underlying graph structure.

## A.2 FURTHER EXPERIMENTAL DETAILS

Let $D$ be the base hidden size of the network. For the standard MLP and master controller, we use a 2 layer neural network with Tanh nonlinearities. The size of the first layer is $D$ while the second layer has $L \cdot D$ hidden units, where $L$ is the number of modules. The standard MLP also has a final linear layer to decode the actions. For all policy architecture variants, the value function is defined as a 2 layer neural network with $D$ hidden units each, followed by a linear layer.

In RL, the loss that all architectures optimize includes the below surrogate objective, a weighted value function loss, and a weighted entropy bonus to encourage exploration:

$$
\begin{aligned}
L^\tau(\theta) &= \mathbb{E}\left[L^\tau_{CLIP}(\theta) - c_1 L^\tau_V(\theta) + c_2 S[\pi_\theta](s^\tau)\right] \\
&= \mathbb{E}\left[\min\left(\hat{A}^\tau r^\tau(\theta), \hat{A}^\tau \mathrm{clip}(r^\tau(\theta), 1 - \epsilon, 1 + \epsilon)\right)\right] \\
&\quad - c_1 \mathbb{E}\left[\left(V_\theta(s^\tau) - V^\tau_{targ}(s^\tau)\right)^2\right] + c_2 \mathbb{E}\left[S[\pi_\theta](s^\tau)\right]
\end{aligned}
\tag{1}
$$

where $\hat{A}^\tau$ is the generalized advantage estimation (GAE) (Schulman et al., 2015). $\epsilon$ is the clip value, $c_1$ is the weight on the value function, and $c_2$ is used to balance the entropy bonus.

We use Adam as the optimizer for both RL and IL. Semicolon-separated entries denote different values for the two domains: "[Locomotion Value]; [Grasping Value]". We conduct an extensive hyperparameter search and find that that the values in Table 1 yield reasonable performance.

For experiments with pretrained policy weights, we initialize the learned logstd to -1.0. For all models on the locomotion tasks, we perform a secondary search over the batch size in [256, 512, 1024, 2048] until performance decays. We find that MeMo works best with smaller batch sizes: 256 on the 12 leg centipede, 512 on the 10 leg worm, and 1024 on the 10 leg hybrid. The hierarchical architecture also sees an improvement when using 1024 for all three robots, whereas the default batch size of 2048 works best for the standard architecture. NerveNet-Conv improves with a batch size of 1024 on the 12 leg centipede and 10 leg worm. We do not see a significant improvement when decreasing the batch size for NerveNet-Snowflake, as the pretrained model overfits to the starting morphology

| Parameters | Value Set |
|---|---|
| Value Loss Factor $c_1$ | 0.5 |
| Entropy Bonus Factor $c_2$ | 0 |
| Discount Factor $\gamma$ | 0.995 |
| GAE $\lambda$ | 0.95 |
| PPO Clip Value $\epsilon$ | 0.2 |
| Gradient Clip Value | 0.5 |
| Starting Learning Rate | 3e-4 |
| Number of Iterations per Update | 10 |
| Learning Rate Scheduler | Decay |
| Number of Processes | 8; 16 |
| Batch Size | 2048; 100 |
| Number of Timesteps | 8e6; 3e5 |
| Base Hidden Size $D$ | 128; 64 |

Table 1: RL Hyperparameters

during imitation learning. Due to the higher variance in reward for grasping tasks, we keep the same batch size for the transfer experiments.

For the hierarchical architectures, we parameterize each joint network within the modules with a 2 layer MLP with 32 hidden units per layer. For imitation learning, we use a batch size of 1024 and tune the learning rate in [7e-4, 1e-3, 2e-3, 4e-3, 7e-3]. All models are trained for 175 iterations of DAgger. We sample 500 trajectories from the expert controllers of the 6 leg centipede and 6 leg worm and 250 trajectories from the controller of the 4 finger claw as the validation sets. We make sure that the architectures pretrained with imitation learning achieve a comparable average reward as the expert controller when a number of trajectories are sampled from them.

For NerveNet, the input network is a single layer with size $D$ followed by a Tanh nonlinearity. We have a separate output network for each joint, and each output network is a 2 layer MLP with 32 units per layer. Table 2 summarizes the hyperparameter search that we perform for NerveNet. We perform grid search over the number of layers and the size of the messages passed by the propagation network. We choose the smallest architecture size that achieves a similar average reward as MeMo. Adding a skip connection from the root to all joints improves NerveNet's validation score in imitation learning and enables the use of smaller architectures that do not overfit as easily.

| Parameters | Value Tried |
|---|---|
| Number of Layers | 2, 3, 4 |
| Message Size | 32, 64, 128 |
| Skip Connection | Yes, No |

Table 2: NerveNet Hyperarameter Search

### A.3 STATE SPACE DESCRIPTION

We keep a running mean and variance to normalize the state space. Relative positions / orientations are relative to a joint in the same module as a given joint. For grasping, we use relative joint orientations as global joint orientations depend significantly on how high the claw is lifted. Table 3 and 4 detail the observation space in locomotion and grasping. For locomotion, "base" refers to the forwardmost wide body segment of the robot. As the joints are hinge joints, they only have one degree of freedom.

### A.4 COMPUTING INFRASTRUCTURE

We run experiments on 2 different machines with AMD Ryzen Threadripper PRO 3995WX processors and NVIDIA RTX A6000 GPUs. Both machines have 64 CPU cores and 128 threads. The main cost of running the agent in both the RoboGrammar and the DiffRedMax environments is the cost of simulation, which is CPU-intensive. For the MLP-based architectures, we only use CPU cores

| Controller Type | Node Type | Observation Type | Axis |
|---|---|---|---|
| master | root | base position | y |
| | | base velocity | x |
| | | base velocity | y |
| | | base velocity | z |
| | | base angular velocity | x |
| | | base angular velocity | y |
| | | base angular velocity | z |
| | | base orientation | x |
| | | base orientation | y |
| | | base orientation | z |
| master, module | joint | joint position | - |
| | | joint velocity | - |
| | | joint orientation | x |
| | | joint orientation | y |
| | | joint orientation | z |
| | | joint relative position | x |
| | | joint relative position | y |
| | | joint relative position | z |

Table 3: Locomotion Observation Space

| Controller Type | Node Type | Observation Type | Axis |
|---|---|---|---|
| master | root | relative fingertip position to object | x |
| | | relative fingertip position to object | y |
| | | relative fingertip position to object | z |
| master, module | joint | joint position | - |
| | | joint velocity | - |
| | | joint relative orientation | x |
| | | joint relative orientation | y |
| | | joint relative orientation | z |
| | | joint relative position | x |
| | | joint relative position | y |
| | | joint relative position | z |

Table 4: Grasping Observation Space

for computing rollouts in parallel environments via vectorization and backpropagating the policy gradient. For NerveNet, in the locomotion domain, we find it helpful to vectorize environments while performing backpropagation with a GPU. For example, the RL stage of MeMo on the 6 leg centipede takes less than a day to complete, whereas training a 3 layer NerveNet-Conv with the same number of processes and batch size requires 3-4 days without a GPU. We note that our resources are shared, and the wallclock time varies depending on the other processes running on the same server.

## A.5 ADDITIONAL ENVIRONMENT DETAILS

We provide more details on the RoboGrammar tasks. On all locomotion tasks, the maximum episode length is 128. Full details of the environments can be found in the RoboGrammar codebase (Zhao et al., 2020).

- Frozen Terrain: A flat surface with a friction coefficient of 0.05.

- Ridged Terrain: Ridges are placed an average of one meter apart across the width of the terrain.

- Gap Terrain: A series of platforms separated by gaps.

- Stepped Terrain: A series of steps with varying height, resembling a flight of stairs.

For all RoboGrammar locomotion environments, the reward at timestep $\tau$ is the sum of the rewards at each sub-step $t$. The training reward function at substep $t$ is

$$R(s_t, a_t) = V_x + 0.1(e_x^{\text{body}} \cdot e_x^{\text{world}} + e_y^{\text{body}} \cdot e_y^{\text{world}}) - 0.7\|a_t\|^2/N \qquad (2)$$

where $N$ is the dimension of the action vector, and each dimension is normalized to [-1, 1]. The first two terms encourage high velocity in the $x$-direction and maintaining the robot's initial orientation respectively. The last term is a regularization penalty to reduce the variance across different runs. The reported reward curves do not include the regularization penalty.

For grasping, the goal is to grasp an object and lift it as high as possible and the maximum episode length is 50. As in prior work (Chen et al., 2021), we follow the convention of controlling the actuators with relative positions rather than absolute positions. The reward at timestep $\tau$ is the sum of the rewards at each sub-step $t$. The full set of parameters used to construct the DiffRedMax simulation will be released with our source code. Below is the reward function used, where $\text{object}_z$ refers to the object's z-coordinate and avg_fingertip_dist is the mean distance of the claw's fingertips to the object's surface. We approximate the cube's surface with the surface of the largest sphere that fits in the cube. all_fingers_in_contact checks whether or not all fingers of the claw is within a small distance from the surface of the object.

$$R(s_t, a_t) = \begin{cases} 10 \cdot \text{object}_z - 0.1 \cdot \text{avg\_fingertip\_dist} & \text{all\_fingers\_in\_contact} \\ -0.1 \cdot \text{avg\_fingertip\_dist} & \text{!all\_fingers\_in\_contact} \end{cases}$$

The penalty on avg_fingertip_dist encourages the fingers to grasp the object. We only include the reward term on $\text{object}_z$ when all_fingers_in_contact is satisfied in order to prevent the claw from throwing the object.

## A.6 SOURCES

We use the PPO implementation provided in `https://github.com/ikostrikov/pytorch-a2c-ppo-acktr-gail`. Our NerveNet implementation is adapted from a PyTorch version of the original NerveNet codebase: `https://github.com/HannesStark/gnn-reinforcement-learning`. We use the official RoboGrammar (Zhao et al., 2020) and DiffRedMax (Xu et al., 2021) simulators.

## A.7 BROADER IMPACT

Here we discuss the broader social impact of our work, including its potential positive and negative aspects. On the positive side, our work enables us to train neural network architectures which are structured in a more interpretable manner, in that the modules correspond to physical components of the robot. In addition, we demonstrate generalization to robot structures and tasks that are greater in difficulty than the training setting. In summary, our modular approach is a step towards addressing the concern that neural networks are black-box models with highly limited generalization capabilities.

We do not see any direct negative implications stemming from our work, as experiments are solely conducted in simulated robot environments. We note that our work does not impose safety constraints on the rollouts of the agent, which is an important limitation to address for real-world use of our method.

## A.8 EXTENDED RELATED WORKS

**Hierarchical RL.** We note that our proposed hierarchical architecture bears similarity to those used in hierarchical RL (Florensa et al., 2017; Hejna et al., 2020). However, a key difference is that our architecture is hierarchical with respect to the morphology of the robot, not the temporal structure of the task.

**Multi-Task RL.** To train robots that perform a diverse set of skills and generalize to new tasks, prior work leverages the shared structure of tasks, such as through graph representations (Li et al., 2022; Kumar et al., 2022) that represent task compositionality, or through language representations (Ahn

et al., 2022; Huang et al., 2022). While many works in MTRL focus on a single morphology, recent efforts (Furuta et al., 2022) have proposed representing both morphology and task in a single graph, enabling architectures trained on this unified IO representation to transfer to unseen morphologies and tasks.

**Multi-Robot Coordination.** Past works in multi-robot coordination bear similarity to our work in either the modularity of the architecture or the learning of a higher-level coordination mechanism, analogous to our master controller, between different agents. In particular (Lee et al., 2020) uses a hierarchical architecture, in which a higher-level meta-policy coordinates various skills with a behavior embedding. Another work (Aljalbout et al., 2023) proposes to learn a useful latent action space for coordinating a multiagent system via an information bottleneck. The information bottleneck helps in learning a latent action space from the full set of observations that is useful in coordinating decentralized agents at inference time.

## A.9 ADDITIONAL EXPERIMENTS

We conduct additional experiments to demonstrate the transfer capabilities of MeMo, particularly with the centipede class of robots. In Fig. 10a, we transfer the 6 leg centipede modules to an 8 leg centipede, which has an additional body module and two leg modules compared to the 6 leg. We demonstrate that MeMo achieves greater than 2x training efficiency compared to all other baselines on the 8 leg centipede. In Fig. 10b, we test the zero-shot generalization of the pretrained NerveNet-Conv baseline by fixing all of its weights and only training the learned logstd when transferring from the 6 to the 12 leg centipede. Its poor performance demonstrates the difficulty of the transfer task, in spite of the physical similarities between the 6 and the 12 leg centipede.

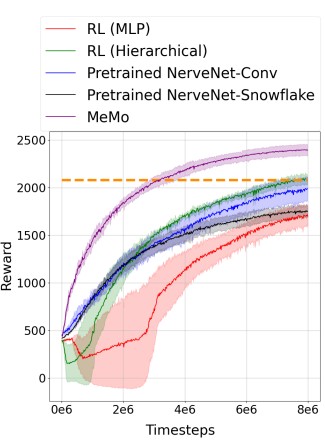
(a) 6 to 8 Leg Centipede Transfer.

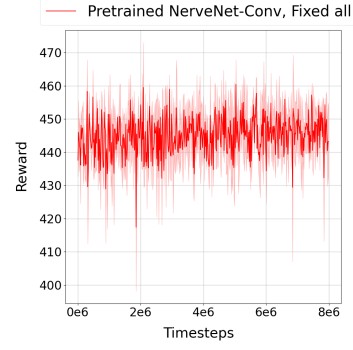
(b) Fixed NerveNet-Conv on 6 to 12 leg Centipede Transfer

Figure 10: **Additional Experiment Results. Left:** 6 to 8 leg centipede transfer on the Frozen Terrain. **Right:** All weights of the NerveNet-Conv baseline, pretrained on the 6 leg centipede, are fixed during transfer to the 12 leg centipede on the Frozen Terrain, resulting in suboptimal performance.

In addition, though transferring control from the 4 to 5 finger claw transfer (Fig. 6) is challenging in the sense that there are more fingers to coordinate, we also demonstrate positive transfer results (Fig. 11) when removing a finger from the starting claw, which can result in a less stable grasp. In Fig. 12, we run MeMo on the 6 to 12 leg centipede transfer where either the master controller or the modules are 4 layer MLPs instead of 2. Both of these variants perform similarly to the original architecture.

## A.10 ADDITIONAL FIGURES

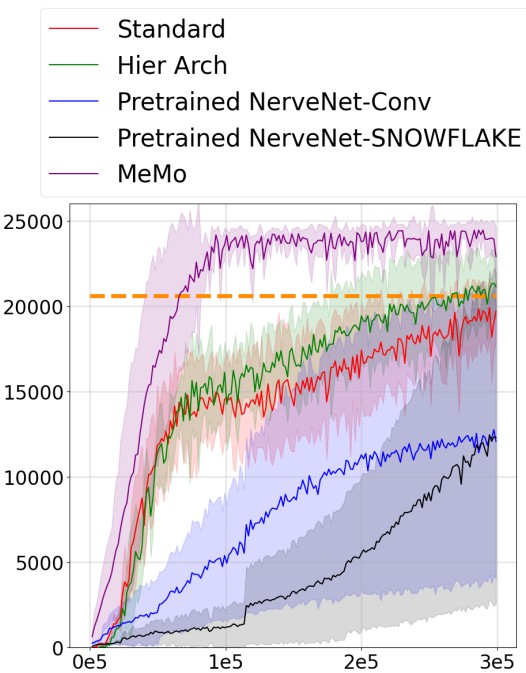

Figure 11: **4 to 3 Finger Claw Transfer Results:** MeMo achieves $3\times$ greater training efficiency compared to the strongest baseline.

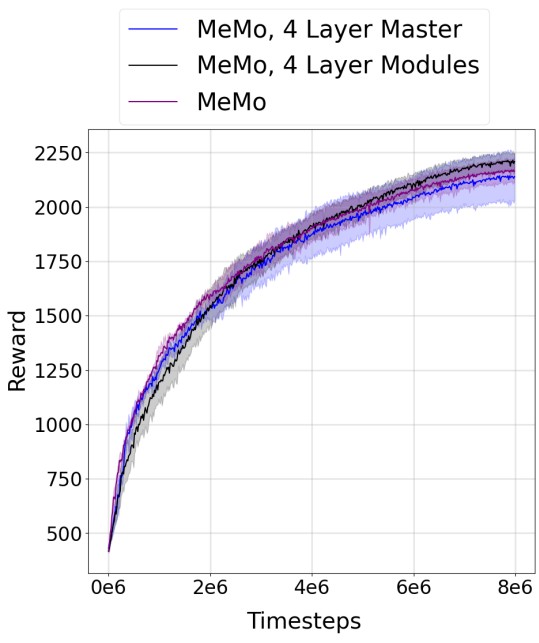

Figure 12: **Architecture Variants of MeMo on 6 to 12 Leg Centipede Transfer:** We run experiments where either the size of the master controller or the size of the modules is increased from 2 to 4 layers. Both of these variants achieve comparable performance to the original architecture with 2 layer MLPs.

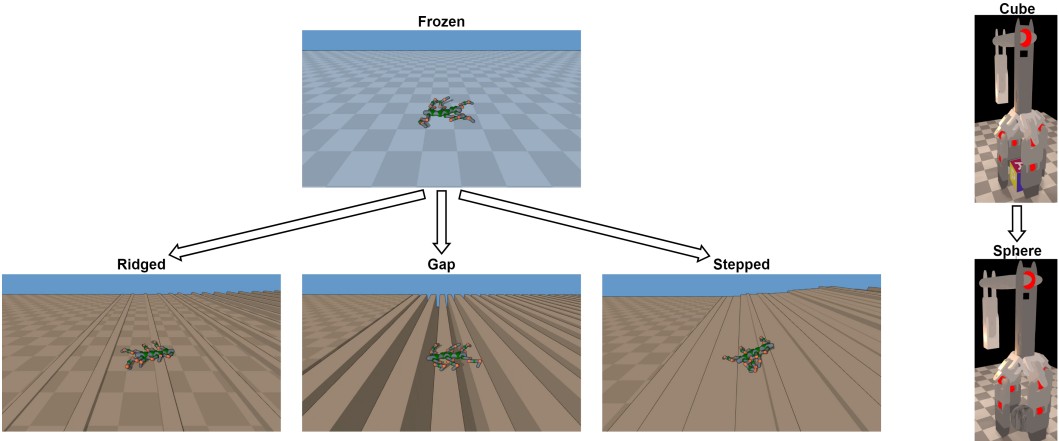

Figure 13: **Task transfer. Left:** The training task is a 6 leg centipede locomoting over the Frozen Terrain. The goal is to transfer policy weights to Ridged, Gap, and Stepped Terrains, all of which require the robot to overcome obstacles unseen in the Frozen Terrain. **Right:** In the grasping domain, the training task is a 4 finger claw lifting a cube, and the testing task is the same claw lifting a sphere. A sphere is naturally a harder object to grasp due to its curved surface.

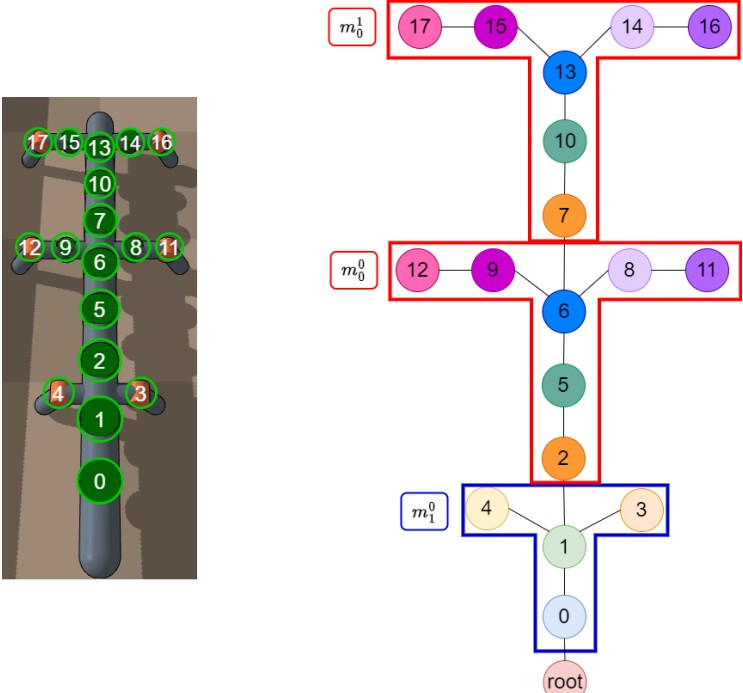

Figure 14: **Graph Structure and Modules of the 6 Leg Worm. Left:** Rendered robot, with joints labeled numerically and circled. **Right:** Corresponding graph structure with joints as nodes and links as edges. The joints circled in red can be thought of the "head" while the joints circled in blue form the "body" modules.

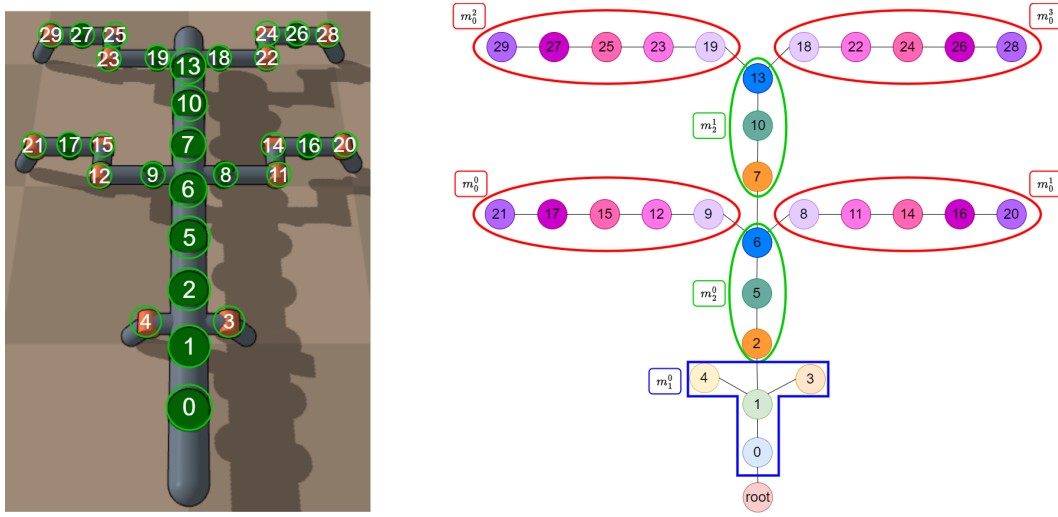

Figure 15: **Graph Structure and Modules of the 6 Leg Hybrid. Left:** Rendered robot, with joints labeled numerically and circled. **Right:** Corresponding graph structure with joints as nodes and links as edges. The joints circled in red belong to the "leg" modules, those circled in green belong to "body" modules, and those circled in blue belong to the "head" module.

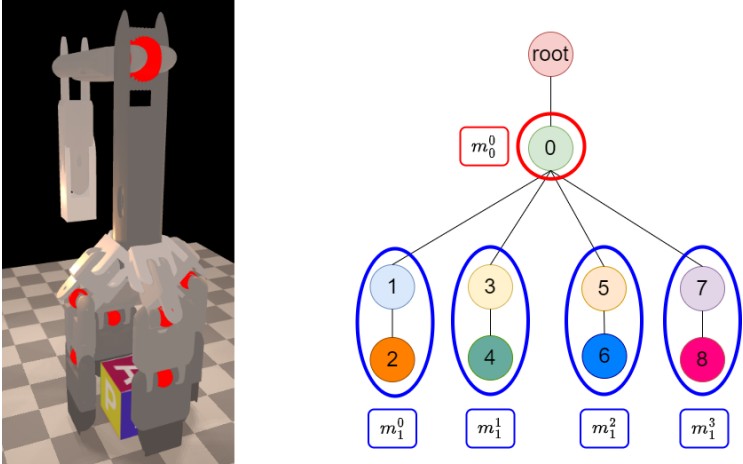

Figure 16: **Graph Structure and Modules of the 4 Finger Claw. Left:** Rendered robot, with joints denoted by red spheres. **Right:** Corresponding graph structure with joints as nodes and links as edges. Each pair of finger joints belongs in its own module, and the arm joint belongs in a separate module.

