# OpenReview forum: "MeMo: Meaningful, Modular Controllers Via Information Bottlenecks"
_ICLR.cc/2024/Conference — Submitted to ICLR 2024_

### Official Review · Reviewer_uVBN · 2023-10-29

**Soundness:** 3 good
**Presentation:** 2 fair
**Contribution:** 2 fair
**Rating:** 5
**Confidence:** 4

**Summary:**

This paper presents MeMo, a hierarchical, modular architecture for robot controllers. It pretrains a modular architecture that assigns separate neural networks to physical substructures and uses an information bottleneck to learn an appropriate division of control information between the modules. The authors have conducted experiments to evaluate the generalization capability of MeMo for both structure and task transfer.

**Strengths:**

1. MeMo introduces an innovative hierarchical architecture for robot control that emphasizes modularity. The introduction of the information bottleneck adds a unique element that aids in generalization.

 2. The paper presents a thorough set of experiments in both structure and task transfer settings. The evaluation covers various robot morphologies and tasks, demonstrating the flexibility and effectiveness of MeMo.

 3. The paper is well-structured, and the key concepts are presented in a clear and understandable manner. Figures and plots help in visualizing the results and concepts.

4. The authors have benchmarked MeMo in challenging environments for robot morphology transfer. The results indicating improved training efficiency in both structure and task transfer are commendable. Moreover, the comparison with other methods, especially NerveNet, provides valuable insights into the advantages of MeMo.

**Weaknesses:**

1. The significance of the information bottleneck is mentioned, but a deeper exploration into its role and implications within the framework might enrich the paper's content. It might be helpful if the authors can clarify the choice of Gaussian noise as the information bottleneck and its specific advantages over other potential techniques. Providing some theoretical underpinnings would enhance the paper's contribution.

 2. While the paper mentions comparisons with other methods like NerveNet, a moreexhaustive comparative analysis might accentuate MeMo's advantages more distinctly. It would be beneficial to provide more granularity on experimental setups, parameter settings, and perhaps even failed experiments to give readers a comprehensive understanding. Please consider extending experiments to more diverse robot morphologies or even real-world robot applications to further validate the framework's robustness.

3. This article is not very innovative to the field.

**Questions:**

1.  I have concerns on extending this method to more complex tasks (environments). The authors should compare their methods with more strong baselines which learn universal controllers for modular robots.

2. Can and how MeMo transfer to the real world problem?

---

> ### Author Response · Authors · 2023-11-17
> **Response to uVBN**
>
> We thank the reviewer for their feedback and are glad that the reviewer finds our empirical evaluation thorough. Below we address the reviewer’s concerns and questions.
>
> **Under "Weaknesses":**
> > The significance of the information bottleneck is mentioned, but a deeper exploration into its role and implications within the framework might enrich the paper's content. It might be helpful if the authors can clarify the choice of Gaussian noise as the information bottleneck and its specific advantages over other potential techniques. Providing some theoretical underpinnings would enhance the paper's contribution.
>
> While we agree that our work would benefit from greater theoretical analysis on the effect of the bottleneck, we believe that there is an intuitive explanation behind the Gaussian noise bottleneck, which we detail in our global response (Part 1). As a result of the added noise, the modules treat the master signal not as a high-dimensional vector but instead as a much lower-dimensional representation, where many of the dimensions have redundant information. Our eigenvalue experiments in Section 4.4 demonstrate that the modules do learn low dimensional representations of the actuator space with respect to the master controller, meaning that the modules themselves must be expressive enough to capture the variance in the actuator space that is not accounted for by the master signal.
>
> > While the paper mentions comparisons with other methods like NerveNet, a moreexhaustive comparative analysis might accentuate MeMo's advantages more distinctly. It would be beneficial to provide more granularity on experimental setups, parameter settings, and perhaps even failed experiments to give readers a comprehensive understanding. Please consider extending experiments to more diverse robot morphologies or even real-world robot applications to further validate the framework's robustness.
>
> In addition to the hyperparameters detailed in Appendix A.2, we have performed additional experiments increasing the number of layers of the master controller and the modules, showing that our framework achieves similar results with deeper MLPs on the 6 to 12 leg centipede transfer. We note that for our framework, we can use most of the same hyperparameters for PPO used to train regular 2 layer MLPs on the same task. We additionally find it sufficient to try smaller batch sizes at transfer time, and we have detailed the specific batch sizes that yield these improvements in Appendix A.2.
>
> > This article is not very innovative to the field.
>
> While many papers [1, 2, 3] have studied the problem of learning modular controllers, these methods take the approach of training on many environments that represent a diverse set of morphologies. However, we focus on an orthogonal dimension of the problem, where we assume that the expert only has a single training robot and wants a fast turnaround when training a new controller for a modified version of the robot, rather than spending significant compute and time to perform multi-task RL. To the best of our knowledge, only NerveNet attempts to do one-shot structure transfer.
>
> [1] Huang, Wenlong et al. “One Policy to Control Them All: Shared Modular Policies for Agent-Agnostic Control.”
> [2] Kurin, Vitaly et al. "My Body is a Cage: the Role of Morphology in Graph-Based Incompatible Control."
> [3] Gupta et al. "MetaMorph: Learning Universal Controllers with Transformers."
>
> **Under "Questions":**
>
> > I have concerns on extending this method to more complex tasks (environments).
>
> The focus of our method is structure transfer, and we see task transfer as an additional benefit on top of the structure transfer we achieve. We agree that because our approach is not specifically designed to tackle task transfer, we only expect for task transfer to be successful when the source and transfer tasks are similar enough. We show the task transfer results as supporting evidence to our structure transfer results that our method does learn meaningful modules. In particular, on task transfer, our method significantly outperforms the hierarchical architecture pretrained end-to-end with RL, which highlights the benefit of the training method we propose.
>
> > Can and how MeMo transfer to the real world problem?
>
> We believe that extending our work to real world problems is a promising future direction and that our experiments have shown evidence that our method has potential to work in real-world applications. Across our examples we have tried a diversity of module shapes that would be commonly found in rigid robots for locomotion and manipulation. As mentioned in our response to reviewer aQSN, visualizations of learned policy’s trajectories show that the modules can have complex coordination mechanics, particularly the behaviors of each leg of the 12 leg centipede.

---

> > ### Comment · Reviewer_uVBN · 2023-11-22
> > **Thank you for your response**
> >
> > Thank you for your response.

---

### Official Review · Reviewer_PL91 · 2023-10-30

**Soundness:** 3 good
**Presentation:** 3 good
**Contribution:** 3 good
**Rating:** 6
**Confidence:** 4

**Summary:**

This paper focuses on learning transferrable control policy for modular robots. To achieve this goal, the authors pretrain a modular architecture that assigns separate neural networks to robot substructures and uses an information bottleneck to learn an appropriate division of control information between the modules. They further benchmark the proposed framework in locomotion and grasping environments. The empirical results validate the effectiveness of the proposed method.

**Strengths:**

1. The problem this paper considers is important.
2. This paper is well-motivated. The idea of reusing control policy for similar modules of robots is easy to understand.
3. The proposed method is generally novel although there might be some works in multi-agent reinforcement learning that have similar network structures.
4. This paper is well-written and easy to follow.
5. The empirical results in Figure 6 are significant compared with NeverNet. But more SOTA baselines should be considered.

**Weaknesses:**

1. The empirical results can be further improved. The authors only compared the proposed method with NerveNet, but ignored any recent works in the field of modular RL, such as SMP (One Policy to Control Them All: Shared Modular Policies for Agent-Agnostic Control), Amorpheus (My Body is a Cage: the Role of Morphology in Graph-Based Incompatible Control), METAMORPH: LEARNING UNIVERSAL CONTROLLERS WITH TRANSFORMERS, etc.
2. The proposed method requires predefining the partition of robots.
3. This paper does not provide any discussions of limitations.

**Questions:**

1. How does the MLP network in Figure 4 handle incompatible input and output?
2. Why does adding Gaussian noise can be regarded as an information bottleneck?

---

> ### Author Response · Authors · 2023-11-17
> **Response to PL91**
>
> We thank the reviewer for their feedback and are glad that the reviewer finds our paper well-written. Below we address the reviewer’s concerns and questions.
>
> **Under "Weaknesses":**
>
> > The empirical results can be further improved. The authors only compared the proposed method with NerveNet, but ignored any recent works in the field of modular RL, such as SMP (One Policy to Control Them All: Shared Modular Policies for Agent-Agnostic Control), Amorpheus (My Body is a Cage: the Role of Morphology in Graph-Based Incompatible Control), METAMORPH: LEARNING UNIVERSAL CONTROLLERS WITH TRANSFORMERS, etc.
>
> Please see our global response (Part 2) for a complete discussion of how our work is related to the literature on learning modular controllers via multi task reinforcement learning. In particular, works that leverage Transformer-based policies (e.g. Amorpheus, MetaMorph) primarily rely on node features, since Transformers disregard connectivity between limbs [1]. As such, they depend on a multi-task setting to generalize, making them fundamentally unsuited to our problem.
>
> [1] Kurin, Vitaly et al. "My Body is a Cage: the Role of Morphology in Graph-Based Incompatible Control."
>
> > The proposed method requires predefining the partition of robots.
>
> We understand that our method’s requirement for a predefined partitioning of joints can be viewed as a limitation. However, we believe that in many cases, it is desirable for experts to inject domain knowledge in our algorithm via their partitioning, which can account for physical constraints. In addition, because our framework only requires the partitioning of joints for a single robot, we do not believe our approach is significantly limited by requirements for expert labels.
>
> > This paper does not provide any discussions of limitations.
>
> Please see our updated paper for a full discussion of the limitations.
>
> **Under "Questions":**
>
> > How does the MLP network in Figure 4 handle incompatible input and output?
>
> $M_1$, $M_2$, and $M_3 $ are all different MLP’s to avoid the incompatibility of the observation spaces seen by $M_2$ and $M_3$. While this requires $M_3$ to be retrained during structure transfer, our experiments show that such retraining is highly sample efficient.
>
> > Why does adding Gaussian noise can be regarded as an information bottleneck?
>
> For a detailed explanation, please see our global response (Part 1). In Section 4.4, with our eigenvalue experiments, we have shown that the modules learn low dimensional representations of the actuator space with respect to the master controller. This means that the modules must be expressive enough to capture the variance in the actuator space that is abstracted away by the master signal.

---

### Official Review · Reviewer_aQSN · 2023-10-31

**Soundness:** 3 good
**Presentation:** 3 good
**Contribution:** 2 fair
**Rating:** 5
**Confidence:** 3

**Summary:**

In this paper, the authors show the benefits of a hierarchical network structure, trained via Imitation Learning and a gaussian-noise induced 'bottleneck', when transferring learned policies between robots with different morphologies. The network is composed of a 'master' network followed by many modules, which correspond to a grouping of similar actuators (one per joint). The authors show that by using the hierarchical network they can achieve sample efficiency and transfer policies between different structures and tasks. The network is compared to hierarchical/non-hierarchical RL agents that starts from scratch as well as to a state-of-the-art transfer approach called Nerve-Net.

**Strengths:**

The authors introduce a very important problem for actual robotics, that of transferring learned policies between robots (with different morphologies, assuming shared actuators/joint structures). The method chosen also looks interesting and several simulation examples are shown to support the claim that the hierarchical network can achieve knowledge/policy transfer between robots (and tasks, although the motivation is for different morphologies).

**Weaknesses:**

However, I found the contribution to be lacking in several key aspects:

* The proposed method was not sufficiently / rigorously analyzed. It is not clear why certain choices were made: the Gaussian-noise induced 'bottleneck' for instance performed better than L1-regularization, but it would be nice to analyze this more mathematically and study how it induces modularity during training.

* It is also not clear why Imitation Learning is necessary to train the hierarchical network with bottleneck, I believe if the authors focus on this problem, it can lead them to more rigorously understand the approach and suggest improvements for RL training.

* It is very difficult to convincingly show knowledge/policy transfer between different 'robots', when these are only in simulation. The robots used in simulation, moreover, have very similar structures, for a more convincing argument I would show more experiments between quite different morphologies and tasks. Even without real robot experiments, it would go a long way to consider several difficulties that show up in real robot experiments: sensor noise, not having access to joint velocity measurements, actuator limits etc.

* It would be nice to show when and if the proposed method fails: this would help to understand how the method contributes to the literature better and also pave the way to more research.

**Questions:**

* Shallow MLPs were used throughout the training as components, have the authors tried other architectures such as convolutional networks, transformers, or deeper MLPs?

* Are there other approaches to compare against besides Nerve-Net? The related work sections mentions many recent papers, some of which could be comparable.

---

> ### Author Response · Authors · 2023-11-17
> **Response to aQSN**
>
> We thank the reviewer for the constructive review and are glad that the reviewer finds our approach interesting. Below we address the reviewer’s concerns and questions.
>
> **Under "Weaknesses":**
>
> > The proposed method was not sufficiently / rigorously analyzed. It is not clear why certain choices were made: the Gaussian-noise induced 'bottleneck' for instance performed better than L1-regularization, but it would be nice to analyze this more mathematically and study how it induces modularity during training.
>
> Please see our global response (Part 1) for a detailed explanation of the intuition behind the Gaussian noise bottleneck. While we agree that our work would benefit from greater theoretical analysis on the effect of the bottleneck, we believe that our results have an intuitive explanation. When noise is added to the master signal, the modules treat the signal as a low-dimensional representation with redundancy. Our eigenvalue experiments empirically demonstrate that the interface between the master and the modules is low-dimensional, forcing the modules to capture the variance in the actuator space that the master controller’s signal abstracts away.
>
> > It is also not clear why Imitation Learning is necessary to train the hierarchical network with bottleneck, I believe if the authors focus on this problem, it can lead them to more rigorously understand the approach and suggest improvements for RL training.
>
> We understand that it may be unintuitive for our approach to succeed with IL but not with RL. However, even with Selective Noise Injection, which improves RL performance by removing noise in rollouts and decreasing gradient variance, the data distribution is non-stationary during RL. Even with an information bottleneck, it is difficult for the modules to learn general features that describe the space of ideal trajectories, since the majority of the training data during RL is produced by an imperfect policy.
>
> > It is very difficult to convincingly show knowledge/policy transfer between different 'robots', when these are only in simulation. The robots used in simulation, moreover, have very similar structures, for a more convincing argument I would show more experiments between quite different morphologies and tasks. Even without real robot experiments, it would go a long way to consider several difficulties that show up in real robot experiments: sensor noise, not having access to joint velocity measurements, actuator limits etc.
>
> As discussed in our global response (Part 1), transfer to diverse morphologies is outside the scope of our work and have revised our Introduction and added a new section on limitations and future work to clarify this point. We note that even in simulation, visualizations of learned policy’s trajectories show that the modules can have complex behavior, particularly the behaviors of each leg of the 12 leg centipede. However, we agree that extending our work to real world robots is a direction for future work.
>
> > It would be nice to show when and if the proposed method fails: this would help to understand how the method contributes to the literature better and also pave the way to more research.
>
> We have updated our paper with an additional section on limitations and directions for future work.
>
> **Under "Questions":**
>
> > Shallow MLPs were used throughout the training as components, have the authors tried other architectures such as convolutional networks, transformers, or deeper MLPs?
>
> We have run additional experiments increasing the master controller and module size from 2 to 4 layers MLPs (Fig. 12). On the 6 to 12 leg centipede transfer, these variants, when trained with our framework, perform comparably to the original architecture.
>
> > Are there other approaches to compare against besides Nerve-Net? The related work sections mentions many recent papers, some of which could be comparable.
>
> Much of the related work explores a fundamentally different problem from our work, of training universal controllers, typically Transformers, over multi-task datasets. As Transformers’ capability to perform tasks rely on attention over node observations, they depend on a multi-task setting to generalize effectively, whereas our single shot transfer necessitates leveraging the modular structure of a single robot as much as possible. Please see our global response (Part 2) for a complete discussion of how our work is related to the literature on learning modular controllers via multi task reinforcement learning.

---

### Official Review · Reviewer_eG93 · 2023-11-01

**Soundness:** 2 fair
**Presentation:** 2 fair
**Contribution:** 2 fair
**Rating:** 3
**Confidence:** 4

**Summary:**

This paper proposes a modular policy learning approach for modularly designed robots. The approach learns separate modules for each actuator of the robot, along with a master controller. The master controller provides higher-level information to each actuator module, which then uses this information along with actuator-specific information to output the torques that are executed at each actuator. The paper uses an information bottleneck technique to learn appropriate modular behaviors. The proposed approach is evaluated on locomotion and manipulation tasks, and demonstrates good performance for both task and morphology transfer.

**Strengths:**

The overall paper focuses on an important problem of learning modular controllers such that they can be transferred from one morphology to another.

**Weaknesses:**

It is really unclear to me why the proposed approach will surely learn modular controllers. Specifically, the approach this paper proposes is to use reinforcement learning to train a monolithic policy (here done via PPO). This learned policy is then used to generate data which is used in an imitation learning setup.

Specifically, the authors say

“The goal of Imitation Learning …. train models that are transferable to variety of context (e.g. robot structures and tasks). This requires modules to represent meaningful behavior, which we enforce by imposing an information bottleneck on the master controller.”

But a fundamental problem with this approach is that the RL policy  has been trained on one task alone. Using an asymmetric approach to train the modular policy (with noise) does not in any way guarantee that the policies learned by individual modules will transfer to new tasks or morphologies. It is only because the task being considered here are too simplistic and almost within domain that we see positive transfer. Specifically, the robot manipulation example uses 4 fingers to train and 5 fingers to evaluate. However, the grasping policy in each of these settings is very similar basically, each finger basically has to push into the object. Further, grasping with greater than 2 fingers is anyways quite easy. Hence, it is really unclear if there is any modular structure. It would be interesting and make the paper much better if more challenging tasks are considered. Specifically, tasks where modularity is well defined.

Can the authors provide any reason why this method will learn truly modular policies? Also, would this approach work if I define each joint of a 7dof robot as a separate module and then try to transfer to a 6-dof robot or another 7-dof robot with very different morphology?

**train models that are transferable to variety of context (e.g. robot structures and tasks)**: Is there any limitation on the kind of tasks that this approach transfers or would this work across all different kinds of tasks? I think these above statements which form the crux of the paper are quite vague and do not precisely tell the reader where this approach can work and what are its failure modes.

**More Manipulation Experiments:** The paper argues that this is a general approach. However, most experiments are in walking domains (e.g. centipedes and worms). Can more manipulation tasks be considered for this approach? For instance, I am wondering whether a task (e.g. opening an oven) when done in slightly different configurations or with objects of different sizes still allows modular controllers to be learned from *a single task instance* (as done here).

**Questions:**

please see above.

---

> ### Author Response · Authors · 2023-11-17
> **Response to eG93, Part 1**
>
> We thank the reviewer for the detailed and constructive review. Below we address the reviewer’s concerns and questions.
>
> **Under "Weaknesses":**
>
> > But a fundamental problem with this approach is that the RL policy has been trained on one task alone. Using an asymmetric approach to train the modular policy (with noise) does not in any way guarantee that the policies learned by individual modules will transfer to new tasks or morphologies. It is only because the task being considered here are too simplistic and almost within domain that we see positive transfer.
>
> We agree that there is no guarantee that the modules generalize broadly across morphologies and tasks that are much different from the one seen at training. We have revised our Introduction to describe the contribution of our work more precisely, that we focus on transferring to incrementally different robots, as domain experts are likely to optimize an existing robot design by making small changes, rather than overhauling the entire design. Even though the changes are incremental, policy transfer is still highly challenging, as shown in our experimental results where neither training policies from scratch or adapting an established GNN baseline performs as well as our framework.
>
> > Specifically, the robot manipulation example uses 4 fingers to train and 5 fingers to evaluate. However, the grasping policy in each of these settings is very similar basically, each finger basically has to push into the object. Further, grasping with greater than 2 fingers is anyways quite easy. Hence, it is really unclear if there is any modular structure.
>
> Grasping and lifting is a modular task in that there are two levels of coordination: first, the fingers need to coordinate with each other to form a stable grasp (one failure mode is that the fingers push in at different times, which we observe in some seeds of the GNN baselines), and second, the arm must lift the object to the full height only once it has been grasped. Our 4 to 5 finger transfer is more difficult in terms of coordinating the fingers. However, as the reviewer suggests, having more fingers can make it easier for the arm to coordinate with the fingers, in that the same “pushing” motion can be adapted from a claw that has already learned to grasp with fewer fingers. In our revised paper, we have added a transfer experiment from a 4 to a 3 finger claw (Fig. 11). Without the support of a fourth finger, it is unclear if the transferred finger modules will converge to a stable grasp with a 3 finger claw. Yet, our experiments still show that MeMo achieves much greater sample efficiency compared to the baselines.
>
> > Can the authors provide any reason why this method will learn truly modular policies?
>
> Please see our global response (Part 1) for a full explanation of how our training pipeline learns an appropriate division of labor between the master controller and the modules. The asymmetry in our architecture prevents the modules from acting independently of the master, while our information bottleneck forces the modules to learn meaningful behavior such that learning to coordinate the modules is easier than learning to control the actuators directly.
>
> > Also, would this approach work if I define each joint of a 7dof robot as a separate module and then try to transfer to a 6-dof robot or another 7-dof robot with very different morphology?
>
> If we define a joint as a separate module, the modules completely rely on the master controller to know how to coordinate with the other joints. In this case, there is no benefit of applying our framework, which is useful for learning expressive modules that capture the inter-variance among joints within the same module, such that learning the coordination between modules much easier.

---

> ### Author Response · Authors · 2023-11-17
> **Response to eG93, Part 2**
>
> > Is there any limitation on the kind of tasks that this approach transfers or would this work across all different kinds of tasks? I think these above statements which form the crux of the paper are quite vague and do not precisely tell the reader where this approach can work and what are its failure modes.
>
> We have included an additional section on the limitations of our work in our updated paper. As described in the global response (Part 2), the main limitations are that 1) we only explore the use of PPO and one type of network architecture, MLP with Tanh nonlinearities, 2) our framework’s capacity for task transfer is limited as we do not explicitly capture task semantics, 3) we focus on transferring to incrementally different morphologies, as when the transfer morphology differs drastically, there is no guarantee that the learned modules represent dynamics that are accurate to the new context.
>
> > The paper argues that this is a general approach. However, most experiments are in walking domains (e.g. centipedes and worms). Can more manipulation tasks be considered for this approach? For instance, I am wondering whether a task (e.g. opening an oven) when done in slightly different configurations or with objects of different sizes still allows modular controllers to be learned from a single task instance (as done here).
>
> In the manipulation domain, we have added a 4 to 3 finger transfer task to demonstrate the usefulness of MeMo in generalizing to more difficult structure transfer tasks. As for experimenting with different configurations or objects of different sizes, our framework is not explicitly designed to support task transfer, since it only captures the modularity in the robot’s physical structure. We believe that extending our framework to incorporate task semantics is an interesting future direction.

---

### Author Response · Authors · 2023-11-17
**Global Response, Part 1**

We thank the reviewers for insightful and constructive reviews. We are glad that the reviewers generally recognize the importance of our problem and that our approach is supported with empirical evidence. In addition to our individual responses to each reviewer, we provide detailed responses to three recurring concerns on the presentation of our method and its evaluation (separated into two parts).

> What is the extent to which we expect MeMo to generalize? What are the limitations of our approach?

From the reviews, we recognize that our paper was not precise enough in its statement of our claimed contribution. Our work is targeted to a specific scenario where a user has a robot with a given morphology and would like to experiment with robots with similar morphologies at minimal cost. This is especially valuable for the kind of modular robots that we evaluate. We claim that our approach can help in such a scenario by producing controller modules that when reused by those similar robots helps them to be trained faster. This is the goal of our paper, and it is what our experiments were designed to evaluate. In our revised paper, we have clarified that our approach assumes a degree of similarity between the training and transfer environments, such that the learned modules’ dynamics are still applicable at transfer time. Our experiments show that this is a highly challenging goal and that existing techniques do not do very well in this scenario.

And while we agree that our work would benefit from a theoretical characterization of why or when our approach works, we demonstrated empirically that the approach accomplishes its goal. In our answer to the second concern, we elaborate on how the observed benefits of MeMo arise from the modularity between the master and the modules, as demonstrated by our eigenvalue experiments, and provide intuition on the effects of the Gaussian noise bottleneck.

Finally, we agree that it would be very valuable to analyze how these results transfer to real robots. It is an exciting direction to explore as future work.

> How is our approach modular? Why do we use the Gaussian noise bottleneck?

Our method’s modularity lies in the division of labor between the master controller and the modules. We seek to strike a balance between two undesirable situations on opposite ends of the spectrum: 1) the modules overfit to the training setup, acting independently of the master, 2) the modules leave much of the control to the master and do not improve training efficiency on transfer. The asymmetry in our architecture addresses the possibility of overfitting by ensuring that the modules do not have enough information about the global state of the robot to act independently from the master. The second concern is addressed by the information bottleneck. As our eigenvalue experiments in Section 4.4 demonstrate, the information bottleneck forces the local controllers to abstract the behavior of the physical assemblies to a small number of degrees of freedom so that learning to use the local controllers is easier than learning to control the actuators directly.

As for the intuition behind the Gaussian noise bottleneck, consider the situation where there is no noise. This allows the master controller to choose how much information to communicate through its latent representation. Once we add a random vector to the representation, the information that the master is communicating is being distorted. In order to properly reconstruct the ground truth trajectories in the presence of noise, the modules react by adding redundancy to the signal, such that the communicated information from the masters is low-dimensional and the modules are able to tolerate error in many dimensions. The robust encoding that the modules learn significantly speed up the training of the master controller because the modules are trained to recover the original low-dimensional communication signal even in the presence of noise, particularly in the early stages of RL training. Our eigenvalue experiments are evidence that the approximate number of degrees of freedom with respect to the master signal is small, and as detailed in Appendix A.2, we are indeed able to use smaller batch sizes when retraining the master controller due to the robust encodings that the modules learn.

---

> ### Author Response · Authors · 2023-11-17
> **Global Response, Part 2**
>
> > Why do we not compare with other approaches that learn distributed controllers that generalize to multiple morphologies and multiple tasks?
>
> As mentioned in Part 1 of our global response, approaches that train on many environments that represent a diverse set of morphologies [1, 2, 3] do not address the problem that our paper aims to solve. That said, we did compare with NerveNet-Snowflake, which uses a GRU for message passing and is the state-of-the-art for high-dimensional control via message passing architectures. We found that in our data-scarce setting, NerveNet-Snowflake is always significantly outperformed on structure transfer by its convolutional counterpart, NerveNet-Conv, which overfits less to the node features due to its limited expressiveness. This illustrates that approaches that work best on a more data-rich setting will not necessarily be ideal for our problem.
>
> [1] Huang, Wenlong et al. “One Policy to Control Them All: Shared Modular Policies for Agent-Agnostic Control.”
> [2] Kurin, Vitaly et al. "My Body is a Cage: the Role of Morphology in Graph-Based Incompatible Control.”
> [3] Gupta et al. "MetaMorph: Learning Universal Controllers with Transformers."
>
> In our revised paper, we updated our Introduction to emphasize that our framework is designed to transfer to incrementally different robot morphologies and removed the first paragraph of our original Introduction, which was misleading. In addition, we have added a Limitation and Future Works section (Appendix A.1) describing that the primary limitations of our work is that  1) we only explore the use of PPO and one type of network architecture, MLP with Tanh nonlinearities, 2) task transfer, as our framework does not explicitly encode task semantics, and 3) transferring to morphologies whose underlying coordinating mechanics are much different from the initial morphology, in which case the modules trained on the starting robot do not accurately capture the dynamics of the transfer robot.
>
> We summarize the changes to our paper below:
> - Revised the Introduction to precisely define the scope of our contribution.
> - Added Appendix A.1 detailing the limitations of our approach and directions for future work.
> - Added a 4 to 3 finger claw experiment (Fig. 11), showing that MeMo achieves transfer benefits even when transferring to a morphology with less stable grasp.
> - Added an ablation study (Fig. 12) analyzing the effects of 1) increasing the size of the modules from 2 to 4 layer network and 2) increasing the size of the master controller. Both variants achieve comparable performance to the original network architecture on the 6 to 12 leg centipede transfer.

---

### Meta-Review · Area_Chair_ytmz · 2023-12-09

**Metareview:**

This paper presents a modular architecture for learning transferrable control policy for modular robots. Information bottlenecks are used to learn an appropriate division of control information between different modules. Empirical results on locomotion and manipulation environments validate the effectiveness of MeMo. The reviewers appreciate the novelty of this work. However, they have raised several concerns such as limited comparisons to recent work in modular policies, requiring predefined partitions, and limited rigor in analyzing failure modes. The rebuttal from the authors have addressed some questions on clarity, but the experimental concerns remain.

**Justification For Why Not Higher Score:**

Limited comparisons to recent work in modular RL.

**Justification For Why Not Lower Score:**

N/A

---

### Decision · Program_Chairs · 2024-01-16

Reject